# High Probability Streaming Lower Bounds for $F_2$ Estimation

## Abstract

A recent paper of Braverman & Zamir (2024) gave a lower bound of $\Omega(\frac{1}{\epsilon^2}\log n)$ for estimating the $F_2$ moment of a stream to within $1 \pm \epsilon$ multiplicative error, resolving the complexity of $F_2$ estimation for constant $\delta$ in the insertion-only model. Using the same techniques, we show that their argument can be adapted to achieve tight $\delta$ dependence. Our key step is to replace the "Exam Set Disjointness" problem with a version that we call "Exam Mostly Set Disjointness." This is the exam version of the Mostly Set Disjointness problem introduced in Kamath et al. (2021).

## 1 Introduction

Data streams present a unique challenge in modern computing. Massive datasets, often arriving at high velocity, necessitate algorithms that process information in a single pass with limited memory. These are known as *streaming algorithms*. For us, a stream consists of increment updates to the coordinates of an underlying frequency vector $x$. This is the insertion-only model of streaming. If decrements are allowed as well, then we refer to this as the turnstile model.

A fundamental task in this setting is the estimation of frequency moments, which summarize the distribution of data items. The $k$-th frequency moment, $F_k$, of a stream with frequency vector $x$ (where $x_i$ is the count of item $i$) is defined as $F_k = \sum_i x_i^k$. The second frequency moment, $F_2$ (also known as the squared $\ell_2$ norm, $\|x\|_2^2$), is particularly important, capturing the skewness of the data and finding applications in database query optimization, network traffic analysis, and machine learning Krishnamurthy et al. (2003); Muthukrishnan (2005); Woodruff et al. (2014).

In the $F_2$ estimation problem, the goal is to approximate $F_2$ within a $(1 \pm \epsilon)$ multiplicative factor, with a success probability of at least $1 - \delta$, using minimal space. The celebrated AMS sketch Alon et al. (1996) provides an upper bound of $O(\frac{1}{\epsilon^2}\log(\frac{1}{\delta})\log n)$ space for streams over a universe of size $n$.

Establishing matching lower bounds has been a long-standing research area. Recent work by Braverman and Zamir Braverman & Zamir (2024) resolved the complexity of $F_2$ estimation for constant failure probability $\delta$ in the insertion-only model, showing a lower bound of $\Omega(\frac{1}{\epsilon^2}\log n)$. However, this result did not capture the optimal dependence on the failure probability $\delta$.

In this work, we adapt the techniques of Braverman and Zamir (2024) to achieve tight dependence on the failure probability $\delta$. Our key contribution is the introduction of the "Exam Mostly Set Disjointness" (EMostlyDISJ) problem, which is a variant of the"Exam Set Disjointness" problem used in Braverman & Zamir (2024). EMostlyDISJ is the "exam" version of the Mostly Set Disjointness problem introduced by Kamath et al. (2021) Kamath et al. (2021). We establish the following main result.

**Theorem 1.** *Let $A$ be a streaming algorithm that, for any data stream of length polynomial in $n$ over a universe of size polynomial in $n$, computes an estimate $\hat{F}_2$ such that $\Pr[|\hat{F}_2 - F_2| \leq \epsilon F_2] \geq 1 - \delta$. For $\epsilon\sqrt{n} \geq \log\frac{1}{\delta}$, the space used by $A$ is at least:*

$$\Omega\left(\log\left(\frac{\epsilon\sqrt{n}}{\log\frac{1}{\delta}}\right)\frac{1}{\epsilon^2}\log\left(\frac{1}{\delta}\right)\right).$$

In the ExamSetDisjointness problem which is a variant of the classical SetDisjointness problem Håstad & Wigderson (2007), each of $n$ players is given a set with elements from some universe. Either (i) all of the sets are disjoint, or (ii) they share a common element, which is also held by a referee. The players communicate sequentially in a one-way fashion and then send a message to the referee who must decide between (i) and (ii).

They show that this implies a lower bound for $F_2$ estimation by a type of direct sum argument. We will follow this argument below, however we refer the reader to Braverman & Zamir (2024) for additional details.

Unfortunately, SetDisjointness does not yield moment estimation lower bounds with the correct $\log \frac{1}{\delta}$ dependence on the failure probability. This inspired Kamath et al. (2021) to introduce a variant called AlmostSetDisjointness where the players must distinguish between their sets being disjoint, and some item occurring in at least half of their sets. This version of the problem allows for moment estimation lower bounds with the correct failure probability dependence.

In this work, we observe that by combining AlmostSetDisjointess with the analysis of Braverman & Zamir (2024), one can obtain a lower bound for $F_2$ estimation in insertion-only streams with optimal dependence on the failure probability.

## 1.1 RESULTS AND OVERVIEW

**Lower bounds for $F_2$ estimation.** We adapt the argument of Braverman & Zamir (2024) to obtain tight failure probability dependence for $F_2$ estimation in the insertion-only model. Specicically, we show

**Theorem 2.** *Let $\mathcal{A}$ be a streaming algorithm that, for any data stream of length polynomial in $n$ over a universe of size polynomial in $n$, computes an estimate $\hat{F}_2$ such that $\Pr[|\hat{F}_2 - F_2| \leq \epsilon F_2] \geq 1n - \delta$. For $\epsilon\sqrt{n} \geq \log \frac{1}{\delta}$, the space used by $\mathcal{A}$ is at least:*

$$\Omega\left(\log\left(\frac{\epsilon\sqrt{n}}{\log\frac{1}{\delta}}\right)\frac{1}{\epsilon^2}\log\left(\frac{1}{\delta}\right)\right).$$

To show this result, we use the Exam Set Disjointness problem of Braverman & Zamir (2024), but replace Set Disjointess with the Mostly Set Disjointess problem of Kamath et al. (2021). Specifically, we conisder the following communication game:

**Definition 1** (Exam Mostly Set Disjointness (EMostlyDISJ)). *The setup involves $t$ players and one referee. Let $U$ be a universe of size $|U|$. The inputs are sets $S_1, \ldots, S_t \subseteq U$ for the players and an element $j \in U$ for the referee. The input sets are promised to be either (i) $M$-almost disjoint or (ii) to have a unique element $j_0 \in U$ that is common to at least $ct$ of the sets for some constant $c \in (0, 1)$. The communication is one-way from player $i$ to player $i + 1$, and finally to the referee. With failure probability at most $\delta$, the referee must decide if the input is an instance of case (i) with the intersecting element $j_0$ is equal to its element $j$.*

We follow the argument of Braverman & Zamir (2024) to show that this communication game requires $\Omega(\frac{m}{t}\log\frac{1}{\delta})$ communication where $m$ is the sum of the sizes of the players' sets and $M = O(\log\frac{1}{\delta})$. Note that in n order to allow for small $\delta$, we allow for a small relaxation of the game. Namely, we allow for a small number of repeated elements.

Given this, we give a lower bound for $F_2$ estimation following the argument of Braverman & Zamir (2024). We briefly describe how the reduction works for $t$ players. Let $n$ be the length of the stream. The stream is then divided into $t$ blocks each of size roughly $n/t$. (In fact the size of each block is $n/(4t)$.) We divide the universe into super-items of size $d = \frac{\epsilon^2 n}{t^2}$, where $n$ is the length of the stream. At the end of the stream, if a super-item is repeated among half of the players, then the $F_2$ of the stream changes by $\Omega(\epsilon n)$ more than in the no instance where no item occurs more than $M = O(\log\frac{1}{\delta})$ times. Combining our lower bound for ExamMostlySetDisjointness with the direct sum argument of Braverman & Zamir (2024), this gives a lower bound of roughly $(\log n)\frac{1}{\epsilon^2}\log(1/\delta)$, for reasonably large $\epsilon$ and $\delta$. We give a more refined bound below, which takes into account that there may be items repeated roughly $\log\frac{1}{\delta}$ times with probability $\delta$ (note that repeats only occur for very small $\delta$, i.e. $\delta < 1/\mathrm{poly}(n)$, since we may choose our universe size $m$ to be poly($n$).

**Bounded and sparse streams.** We consider two classes of insertion-only streams for which the lower bound above does not apply. In fact, in these cases, we show that it is possible to beat the lower bound with a better algorithm.

The first situation is streams with entries bounded by $B$. The main observation is that when $B$ is small, we obtain a good estimate of $F_2(x)$ by subsampling entries from the support of $x$. This reduces the size of the vector dramatically, and so the AMS sketch of Alon et al. (1996) has lower bit complexity per entries, reducing the dependence on $\log n$ to roughly $\log B$. A full statement is given in Theorem 4. There are several complications in following the outlined procedure. In particular the subsampling rate needs to depend on the support size, which we do not have access to at the start of the stream. To handle this, we use a continuous $F_0$ tracker, to estimate the sparsity at all points in time.

We also consider sparse streams, where we can first hash the universe items into buckets, while approximately preserving $F_2$ since there are few collisions if we use enough buckets. We then compose with an AMS sketch on the resulting compressed vector. To save space, we do not store the entries of the AMS sketch exactly, but rather use Morris Counters Morris (1978) to separately estimate the positive and negative contributions to each bucket. This roughly allows us to replace the $\log n$ dependence with $\log k$, where $k$ is the sparsity. A full statement is given in Theorem 5.

## 2 INSERTION-ONLY LOWER BOUNDS FOR $F_2$ ESTIMATION.

We present a strengthened space lower bound for the problem of $F_2$ estimation, which is the problem of returning a $(1 \pm \epsilon)$-approximation to $\|v\|_2^2$ with failure probability $\delta$ in the insertion-only model. The proof largely follows the ideas of Braverman & Zamir (2024), but substitutes the underlying Set Disjointness communication problem with a variant called Mostly Set Disjointness, which is a problem introduced in Kamath et al. (2021). Our improvement is the incorporation of the failure probability $\delta$ into the lower bound, namely, we will show:

**Theorem 3.** *Let $\mathcal{A}$ be a streaming algorithm that, for any data stream of length polynomial in $n$ over a universe of size polynomial in $n$, computes an estimate $\hat{F}_2$ such that $\Pr[|\hat{F}_2 - F_2| \le \epsilon F_2] \ge 1n - \delta$. For $\epsilon\sqrt{n} \ge \log\frac{1}{\delta}$, the space used by $\mathcal{A}$ is at least:*

$$\Omega\left(\log\left(\frac{\epsilon\sqrt{n}}{\log\frac{1}{\delta}}\right)\frac{1}{\epsilon^2}\log\left(\frac{1}{\delta}\right)\right)$$

For a communication protocol $\Pi$ with inputs $X$ drawn from a distribution $\mu$ and public randomness $P$, the conditional information cost is the mutual information $I(X; \Pi(X)|P)$ between the input and the protocol transcript. The communication cost of any protocol is an upper bound on its information cost. For a streaming algorithm processing a random stream $X$, we consider the information $I(X; M)$ that the memory state $M$ contains about the stream. We first define the communication problem at the center of our reduction.

In the following definition we will say that sets $S_1, \ldots, S_t$ are $M$-almost-disjoint if

$$\#\{(i, x) : x \in S_i \text{ and } x \in S_j \text{ for some } j \ne i\} \le M.$$

This is a minor technical relaxation of disjointness that we will use to allow for negligible amounts of overlap among the sets.

**Definition 2** (Exam Mostly Set Disjointness (EMostlyDISJ))**.** *The setup involves $t$ players and one referee. Let $U$ be a universe of size $|U|$. The inputs are sets $S_1, \ldots, S_t \subseteq U$ for the players and an element $j \in U$ for the referee. The input sets are promised to be either (i) $M$-almost disjoint or (ii) to have a unique element $j_0 \in U$ that is common to at least $ct$ of the sets for some constant $c \in (0, 1)$. The communication is one-way from player $i$ to player $i + 1$, and finally to the referee. With failure probability at most $\delta$, the referee must decide if the input is an instance of case (i) with the intersecting element $j_0$ is equal to its element $j$.*

By a standard direct sum argument, the conditional information cost of the problem on universe $U$ will be at least $|U|$ times the cost of a 1-bit version, which we call the $F_{ct,t}$ problem. In the $F_{ct,t}$ problem, each of $t$ players receive bits $Y_i \in \{0, 1\}$ and must distinguish between the Hamming weight of $Y$ being at most 1 from the Hamming weight of $Y$ being at least $ct$. Our first goal is to lower-bound the conditional information cost of a protocol for $F_{ct,t}$.

**Definition 3** (Hard Distribution $\nu$). *Let $P$ be a public random variable uniform in $[t] = \{1, 2, \ldots, t\}$. The distribution $\nu$ over inputs $Y \in \{0,1\}^t$ is defined as follows: conditioned on $P = i$, the input $Y$ is the all-zeros vector $0^t$ with probability $1/2$ and the standard basis vector $e_i$ with probability $1/2$. All inputs drawn from $\nu$ are valid NO instances for $F_{ct,t}$.*

We note that $\nu$ is the distribution $\mu_0$ in the notation of Kamath et al. (2021).

**Lemma 1** (Conditional Information Cost on $\nu$, see Kamath et al. (2021)). . *Let $\Pi$ be a protocol that solves $F_{ct,t}$ with failure probability $\delta$. Let $Y$ be drawn from the distribution $\nu$ above. The conditional information cost with respect to the NO-instance distribution $\nu$ is lower-bounded by:*

$$I(Y; \Pi | P) \geq \Omega\left(\frac{1}{t}\log\left(\frac{1}{\delta}\right)\right)$$

*Proof.* Lemma 3.7 of Kamath et al. (2021) shows a conditional information cost lower bound of

$$I(Y; \Pi | P) \geq \Omega(1),$$

as long as $\delta$ is small enough so that $t \leq c\log(\frac{1}{2e\delta})$. Similar to the later argument of Kamath et al. (2021), we boost the failure probability to get a bound for larger $\delta$.

Now suppose we have a protocol $\Pi$ with failure probability $\delta_1$ (which does not necessarily satisfy the constraint above). We boost this protocol to construct a protocol $\Pi'$ with smaller failure probability. Specifically, consider running the protocol $r$ times and taking a majority vote. Then the probability of failure for $\Pi'$ is at most

$$\frac{2r}{r}\delta_1^{r/2} \leq (4\delta_1)^{r/2}.$$

So by taking $r = \frac{ct}{\log\frac{1}{\delta}}$, we ensure that the failure probability satisfies the bound above. This implies that

$$\Omega(1) = I(\Pi^1, \ldots, \Pi^r; Y | D) = \sum_i I(\Pi^i; Y | D, \Pi^{<i}),$$

so by averaging there exists an $i$ for which

$$I(\Pi^i; Y | \Pi^{<i}, D) = \Omega(1/r).$$

But

$$I(\Pi^i; Y | \Pi^{<i}, D) = H(\Pi^i | \Pi^{<i}, D) - H(\Pi^i | \Pi^{<i}, D, Y) = H(\Pi^i | \Pi^{<i}, D) - H(\Pi^i | D, Y)$$

since $\Pi^i$ is independent of $\Pi^{<i}$ conditioned on $D$ and the inputs. Further, $H(\Pi^i | \Pi^{<i}, D) \leq H(\Pi^i | D)$ since conditioning cannot increase entropy. Thus, $I(\Pi^i; Y | D) = H(\Pi^i | D) - H(\Pi^i | D, Y) = \Omega(1/r) = \Omega(\frac{1}{t}\log\frac{1}{\delta})$ and $\Pi^i$ is just our base protocol $\Pi$.

$\square$

We now use the technique of Braverman & Zamir (2024) to relate the conditional information cost on $\nu$ to the information cost on the distribution $\mu_p$, where each bit is $1$ with probability $p$.

**Lemma 2** (Braverman & Zamir (2024) Lemma 4.3, adapted for $F_{ct,t}$). *Let $\Pi$ be a communication protocol for $F_{ct,t}$ that is correct on all inputs satisfying the promise Let $p < 1/t$. For an input distribution $\mu_p$ where each bit is $1$ independently with probability $p$, the information cost is $I_{\mu_p}(Y; \Pi) \geq \Omega(p\log(1/\delta))$.*

*Proof.* The proof follows the structure of Lemma 4.3 in Braverman & Zamir (2024). First, we relate the unconditional and conditional information costs for the distribution $\nu$ for $P$ as defined above. By the chain rule for mutual information: $I(Y, P; \Pi) = I(Y; \Pi) + I(P; \Pi | Y) = I(Y; \Pi)$, since $I(P; \Pi | Y) = 0$. Also, $I(Y, P; \Pi) = I(P; \Pi) + I(Y; \Pi | P) \geq I(Y; \Pi | P)$. Therefore, $I(Y; \Pi) \geq I(Y; \Pi | P)$.

Using the chain rule for mutual information and the derivation involving $D$ as in the proof of Lemma 4.3 in Braverman & Zamir (2024), we have $I_{\mu_p}(Y; \Pi) \geq \Theta(p \cdot t) \cdot I_\nu(Y; \Pi)$. Combining with the above and Lemma 1:

$$I_{\mu_p}(Y; \Pi) \geq \Theta(p \cdot t) \cdot I(Y; \Pi | P) \geq \Theta(p \cdot t) \cdot \Omega\left(\frac{1}{t}\log\left(\frac{1}{\delta}\right)\right) = \Omega\left(p\log\left(\frac{1}{\delta}\right)\right).$$

$\square$

We now apply the reduction from an $m$-element problem to $|U|$ parallel 1-bit instances, as in Section 4.2 of Braverman & Zamir (2024).

We also note the following simple fact, which will show that accidental duplications do not affect the $F_2$ substantially.

**Proposition 1.** *Let* $p = m/(2t|U|)$ *as above. Suppose that each player samples* $m/t$ *elements uniformly from U. Let* $x_i^j$ *be the ith item sampled by player j. Then with probability at least* $1 - \delta$,

$$\#\{(i,j) : x_i^j \text{ sampled by another player}\} \leq c \log \frac{1}{\delta}.$$

*Proof.* The probability that a given set of pairs $(i,j)$, of size $c \log \frac{1}{\delta}$, all contain repetitions of some other element is at most $(\frac{1}{m^2})^{-c \log \frac{1}{\delta}}$ by our choice of $|U|$. There are at most $\binom{m}{c \log \frac{1}{\delta}} \leq \frac{em}{c \log \frac{1}{\delta}}^{c \log \frac{1}{\delta}}$ ways to choose such a set of pairs, so the claim follows by a union bound. $\square$

**Corollary 1.** *Suppose that* $\delta \leq t \exp^{-m/(2t)}$ *. The information cost of an* $O(\delta)$*-error protocol for* $t$*-party EMostlyDISJ with* $M = c \log \frac{1}{\delta}$*, total set size* $m$*, over universe U of size at least* $m^4$ *is*

$$\Omega \left( \min(m, \frac{m}{t} \log \left( \frac{1}{\delta} \right)) \right).$$

*Proof.* The reduction creates player sets from $|U|$ parallel instances of the 1-bit problem. The inputs for these instances are drawn i.i.d. from $\mu_p$, where $p = m/(2t|U|)$. Following the proof of Lemma 4.4 and Lemma 4.5 of Braverman & Zamir (2024), the total information cost is the sum over the $|U|$ parallel instances, which gives:

$$I(X'; \Pi) \geq \sum_{j=1}^{|U|} I_{\mu_p}(Y^j; \Pi) \geq |U| \cdot \Omega(p \log(1/\delta))$$

Substituting $p = m/(2t|U|)$ gives a total cost of:

$$|U| \cdot \Omega \left( \frac{m}{2t|U|} \log(1/\delta) \right) = \Omega \left( \frac{m}{t} \log(1/\delta) \right).$$

Note that the proof of Lemma 4.5 of Braverman & Zamir (2024) has a failure event, which is that one of the players' sets is larger than $m/t$. However this only happens with probability $t \exp(-m/(2t)) \leq \delta$ by Lemma 4.4 of Braverman & Zamir (2024), and our assumption. Finally, the conclusion of the previous proposition holds with failure probability at most $\delta$. $\square$

## 2.1 $F_2$ ESTIMATION LOWER BOUND.

We now follow the structure of Section 5 in Braverman & Zamir (2024), with essentially no changes. For the multi-scale argument, we use a reduction parameterized by $t \leq \epsilon \sqrt{n}$, $M \leq O(\log \frac{1}{\delta})$ (our almost-disjointness parameter), and $d := \lfloor \frac{\epsilon^2 n}{t^2} \rfloor$ (the super-element size). The EMostlyDISJ instance here has a total set size parameter of $m = n/d = \Theta(t^2/\epsilon^2)$.

We construct a stream of length $n$ encoding EMostlyDISJ instances at multiple scales. As in the proof of Braverman & Zamir (2024) we can have at most $\epsilon \sqrt{n}$ players or the super-element size becomes smaller than 1. We choose levels in powers of 2 as in Braverman & Zamir (2024), and the number of such levels is $\Omega(\log n)$ given that $\frac{\epsilon \sqrt{n}}{\log(1/\delta)} = n^{\Omega(1)}$.

For each level $l$, we set $t = 2^l$. The stream is partitioned into "active buckets" for each level's problem. The rest of the stream is filled with elements drawn i.i.d. from a uniform distribution over $U$. As in Braverman & Zamir (2024), we analyze the performance of a streaming algorithm on this random stream distribution, which is identical to a stream where every element is drawn i.i.d. from $U$ given that the universe size is at least $m^4$ and so the $F_2$ of the resulting stream changes by at most a constant factor with failure probability at most $\delta$.

At each level, there are at most $M = \log \frac{1}{\delta}$ repeated items, which contribute at most $\log^2(1/\delta)d$ to $F_2$. As long as $t \geq \sqrt{\epsilon} \log \frac{1}{\delta}$, this is at most an additive $\epsilon F_2$ contribution to the overall $F_2$.

Given the above, say that a level $\ell$ is good if for that level, $t = 2^\ell/4$ satisfies $t \geq \log \frac{c}{\delta}$ (and consequently $\delta \geq \exp(-m/(2t))$), and also $t \leq \epsilon\sqrt{n}$. The former condition guarantees a lower bound of $\frac{t}{\epsilon^2} \log \frac{1}{\delta}$ at each level, and the latter condition guarantees that our super-item size is at least one. We assume that $\log \frac{1}{\delta} \leq \sqrt{n}$ so that there are still $\Theta(\log n)$ good levels.

As we have seen, the information complexity of a single level-$l$ communication phase, where the protocol is an execution of the streaming algorithm over the corresponding stream segment, is lower-bounded by:

$$\Omega\left(\frac{t}{\epsilon^2} \log\left(\frac{1}{\delta}\right)\right),$$

for a good level. This follows directly from the definition of the reduction and our bounds for EMostlyDisj above.

Now we observe that an $F_2$ estimation scheme can solve EMostlyDisj via the current reduction. Faced with a configuration corresponding to a NO instance of our exam-mostly-set-disjointness problem there is a superitem that is repeated $t/2$ times. So if we append $k = t/\epsilon$ copies of the super-item to the end of the stream, this increases the $F_2$ by at least $(k + t/2)^2 - (t/2)^2$. On the other hand, in a NO instance, this increases the $F_2$ of the stream by at most $(k + \log \frac{1}{\delta})^2 - \log^2 \frac{1}{\delta}$. The former is at least $\epsilon n$, and the latter is at most $\frac{\epsilon}{2} n$, by our assumption that $t \geq c \log \frac{1}{\delta}$.

**Definition 4.** *As in [Braverman & Zamir (2024)](#), for a level $l$, with $t = 2^l$, let*

$$I_l := \sum_{j=1}^{n} I\left(X_{(j-\frac{n}{t}, j]}; M_j \mid M_{j-\frac{n}{t}}\right).$$

**Lemma 3** (Essentially Lemma 5.7 of [Braverman & Zamir (2024)](#)). *For each good level $l$, we have $I_l \geq \Omega\left(\frac{n}{\epsilon^2} \log\left(\frac{1}{\delta}\right)\right)$.*

*Proof.* The proof is identical to Lemma 5.7 of [Braverman & Zamir (2024)](#), with an extra $\log \frac{1}{\delta}$ factor throughout. □

As in [Braverman & Zamir (2024)](#), let $\bar{I} := \frac{1}{n} \sum_{j=1}^{n} I(X_{<j}; M_j)$. We also have the following fact from [Braverman & Zamir (2024)](#)

**Lemma 4** ([Braverman & Zamir (2024)](#) Lemma 5.10). *$\bar{I} \geq \frac{1}{n} \sum_\ell I_l$.*

*Proof of Theorem 3.* By combining Lemma 3 and Lemma 4:

$$\bar{I} \geq \frac{1}{n} \sum_\ell I_l$$

$$\geq \frac{1}{n} \sum_\ell \Omega\left(\frac{n}{\epsilon^2} \log\left(\frac{1}{\delta}\right)\right)$$

$$= \sum_\ell \Omega\left(\frac{1}{\epsilon^2} \log\left(\frac{1}{\delta}\right)\right),$$

where the sum is over good levels. The number of terms in the summation is $\Omega(\log \frac{\epsilon\sqrt{n}}{\log \frac{1}{\delta}})$. Therefore,

$$\bar{I} \geq \Omega(\log \frac{\epsilon\sqrt{n}}{\log \frac{1}{\delta}}) \cdot \Omega\left(\frac{1}{\epsilon^2} \log\left(\frac{1}{\delta}\right)\right)$$

The space $M$ of the algorithm must be at least the average information it stores, $M \geq \bar{I}$. □

## 3 BOUNDED AND SPARSE FREQUENCIES

We present an $\ell_2$-estimation algorithm for insertion-only streams where the frequencies are bounded by $B$. The goal is to achieve a space complexity where the dependence on $B$ is only logarithmic, and the dependence on $n$ is also logarithmic, independent of $1/\varepsilon^2$.

**Theorem 4.** *Let $x \in \mathbb{Z}_{\geq 0}^n$ be the frequency vector of an insertion-only stream of length $m$. Suppose that $0 \leq x_i \leq B$ for all $i \in [n]$. Then there is a streaming algorithm that computes a $(1 \pm \varepsilon)$-approximation to $\|x\|_2^2$ with probability $1 - \delta$ using*

$$O\left(\frac{1}{\varepsilon^2} \log^2\left(\frac{B}{\epsilon}\right) \log\left(\frac{1}{\delta}\right) \left(\log B + \log\left(\frac{1}{\varepsilon}\right)\right) + \log n \cdot polylog(n, B, 1/\delta)\right)$$

*bits of space.*

*Proof.* The strategy is to use subsampling to reduce the effective dimension of the problem to a size that depends polynomially on $B$ and $1/\varepsilon$, but only logarithmically on $n$. Then, we apply an efficient $\ell_2$ estimator on this reduced stream.

We use a pairwise independent hash function $h : [n] \to [2^L]$ (where $L = O(\log n)$). We define levels of sampling based on the number of trailing zeros in the hash value $h(i)$. Let $z(i)$ be the number of trailing zeros of $h(i)$. We aim to find a sampling level $\ell$ such that the number of distinct items $i$ with $z(i) \geq \ell$ is approximately $K$, where $K = O(\frac{B^2}{\varepsilon^2} \log(1/\delta))$.

We first utilize an $F_0$ estimation algorithm Kane et al. (2010) to estimate the total number of distinct elements $F_0$. By setting the failure probability of the $F_0$ estimator to $\delta / \log n$, we may assume that the $F_0$-estimator yields a constant factor approximation to $F_0$ at all times throughout the stream (since the stream is insertion-only it only need be correct at checkpoints where the $F_0$ changes by a constant factor). Let $F_{0,t}$ be the number of nonzero entries at time $t$ of the stream, and let $\hat{F}_{0,t}$ be the corresponding estimator, which is a constant factor approximation of $F_{0,t}$. We first describe a two pass version of our algorithm, and then in the following paragraph explain how to extend it to a single pass.

**A two-pass algorithm** On the first pass we compute $\hat{F}_0$. We choose the sampling level $\ell$ such that $2^\ell \approx \hat{F}_0/K$. This ensures that the expected number of items surviving the sampling is within a constant factor of $K$. In other words, let $S$ be the set of surviving coordinates: $S = \{i \in [n] \mid z(i) \geq \ell\}$. The expected size of $S$ is $F_0/2^\ell \approx K$.

We define the natural estimator based on the sampled set $S$: $Y = 2^\ell \sum_{i \in S} x_i^2$.

Let $p = 2^{-\ell}$, so that $Y = \frac{1}{p} \sum_{i \in S} x_i^2$. Since $Y$ is a sampling estimator, we have $\mathbb{E}[Y] = \|x\|_2^2$, and $\text{Var}(Y) = \frac{1-p}{p} \|x\|_4^4 \leq \frac{1}{p} \|x\|_4^4$. Since $x_i \leq B$, $\|x\|_4^4 \leq B^2 \|x\|_2^2$. $\text{Var}(Y) \leq \frac{B^2}{p} \|x\|_2^2$.

To achieve our guarantee it suffices to have $\text{Var}(Y) \leq \varepsilon^2 \|x\|_2^4$. This holds if $p \geq \frac{B^2}{\varepsilon^2 \|x\|_2^2}$. Since $\|x\|_2^2 \geq F_0$, it suffices to have $p \geq \Omega(\frac{B^2}{\varepsilon^2 F_0})$. Our choice of $\ell$ (and thus $p$) ensures this, assuming the constants are set appropriately.

### Implementation and Space Complexity

We need to implement the estimation of $Y$ in the streaming model. We do not explicitly store the set $S$. Instead, we observe that the stream restricted to the coordinates in $S$ is itself a stream. Let $x|_S$ be this restricted vector. We want to estimate $\|x|_S\|_2^2$.

The vector $x|_S$ has an expected dimension of $K$. The frequencies are still bounded by $B$. The $\ell_1$ norm of $x|_S$ is $\|x|_S\|_1 \leq B|S|$. In expectation, $\|x|_S\|_1 \leq BK$.

We apply the standard AMS sketch to the substream corresponding to $S$. When an update $(i, \Delta)$ arrives, we check if $z(i) \geq \ell$. If so, we update the AMS sketch.

The AMS sketch requires $O(\frac{1}{\varepsilon^2} \log(1/\delta))$ counters. The maximum value of a counter is bounded by $\|x|_S\|_1$. With high probability (using Markov's inequality and concentration bounds on $|S|$), $\|x|_S\|_1 = O(BK)$.

The space required for the AMS sketch on the substream is:

$$O\left(\frac{1}{\varepsilon^2} \log\left(\frac{1}{\delta}\right) \log(BK)\right).$$

Substituting $K = O(\frac{B^2}{\varepsilon^2} \log(1/\delta))$:

$$O\left(\frac{1}{\varepsilon^2} \log\left(\frac{1}{\delta}\right) \log\left(\frac{B^3}{\varepsilon^2} \log(1/\delta)\right)\right) = O\left(\frac{1}{\varepsilon^2} \log\left(\frac{1}{\delta}\right) (\log B + \log(1/\varepsilon) + \log\log(1/\delta))\right).$$

We also need space for the $F_0$ estimator and the hash function $h$. The $F_0$ estimator uses $O(\log n \cdot \mathrm{polylog}(n, 1/\delta))$ space (to ensure high probability success throughout the stream). The hash function $h$ uses $O(\log n)$ space.

The total space complexity is dominated by the AMS sketch on the subsampled stream and the $F_0$ estimator:

$$O\left(\frac{1}{\varepsilon^2} \cdot \mathrm{polylog}(B, 1/\varepsilon, 1/\delta) + \log n \cdot \mathrm{polylog}(n, 1/\delta)\right).$$

This satisfies the requirement that the dependence on $n$ and $B$ is logarithmic, and the main factor is $1/\varepsilon^2$.

**Extension to a single pass.** To extend to a single pass, one could run $O(\log n)$ instances of the above single pass algorithm in parallel, using one $p$ at every scale. Then at the end of the algorithm, we have access to $\hat{F}_0$ and can use the instance that chose the appropriate value for $p$. This would increase the overall space by a $\log n$ factor. To do better, we use our continuous $F_0$ tracker. Let $t_0 < t_1 < \ldots$ be the first times for which $\widehat{F_{0,t_i}} \geq 2^i$. At each time $t_i$ we create $N = \lceil \log \frac{B^2}{\epsilon^2} \rceil$ new instances of our single pass sketch using the values of $p$ corresponding to $F_0$ estimates of $2^0 \widehat{F_{0,t_i}}, \ldots 2^{N-1} \widehat{F_{0,t_i}}$. Moreover we only maintain this collection of sketches corresponding to times $t_i, \ldots, t_{i-N}$, so in total this increases our space by a factor of $N^2 = O(\log^2 \frac{B}{\epsilon})$.

Now at the end of the algorithm, we use one of the $N$ sketches in the from the earliest remaining group of non-discarded sketches. Among the $N$ sketches in this group, we choose the one that used the correct value of $p$ given our estimate of $F_0$. This guarantees a $(1 \pm \epsilon)$ approximation to $F_2$ for a suffix of the stream $x_2$. Let $x_1$ be the updates from the corresponding prefix so that $x = x_1 + x_2$. Then by our construction $\|x_1\|_0 \leq \frac{\epsilon^2}{B^2} \|x\|_0$, which by the bounded entry assumption, and the fact that all nonzero entries are at least one, implies that $\|x_1\|_2 \leq \epsilon \|x\|_2$. By the triangle inequality, $\|x\|_2$ and $\|x_2\|_2$ are within $\epsilon \|x\|_2$ of one another. Adjusting $\epsilon$ by a constant factor then yields the desired guarantee. $\qquad\square$

### 3.1 An $\ell_2$-Estimation Algorithm for Sparse Streams

We present an $\ell_2$-estimation algorithm for insertion-only streams where the frequency vector $x$ is $k$-sparse ($\|x\|_0 \leq k$). This algorithm achieves space complexity where $k$ appears only in logarithmic factors. The idea is that we can first hash the coordinates of $x$ into buckets to reduce the dimensionality of $x$. Then we run AMS sketch on the resulting vector, using Morris counters to keep approximate counts of the positive and negative components to each coordinate of the AMS sketch.

**Theorem 5.** *Let $x \in \mathbb{Z}_{\geq 0}^n$ be the frequency vector of an insertion-only stream of length $m$. Suppose that $\|x\|_0 \leq k$. Then there is a streaming algorithm that computes a $(1 \pm \varepsilon)$-approximation to $\|x\|_2^2$ with probability $1 - \delta$ using a total space of*

$$O\left(\frac{1}{\varepsilon^2} \log\left(\frac{1}{\delta}\right) \left(\log\left(\frac{k}{\varepsilon}\right) + \log\log m\right) + \log n\right).$$

*Proof.* The strategy involves dimensionality reduction tailored for sparse vectors, followed by applying the AMS sketch on the reduced vector, where the sketch counters are maintained approximately using Morris counters.

We first reduce the dimension by hashing coordinates of $x$ into random buckets. Let $M = C(k/\varepsilon^2)$ for a sufficiently large constant $C$. We use a pairwise independent hash function $h : [n] \to [M]$, requiring $O(\log n)$ space. We define the reduced vector $y \in \mathbb{R}^M$ where $y_j = \sum_{i:h(i)=j} x_i$. Since the stream is insertion-only, $y_j \geq 0$. Due to the $k$-sparsity of $x$, with constant probability, $\|y\|_2^2 = (1 \pm \varepsilon/4)\|x\|_2^2$.

We apply the AMS sketch to the vector $y$. We use $R = O(1/\varepsilon^2)$ estimators and $T = O(\log(1/\delta))$ repetitions. Let $A_{r,t}$ be the counters:

$$A_{r,t} = \sum_{j=1}^{M} \sigma_{r,t}(j) y_j,$$

where $\sigma_{r,t} : [M] \to \{-1, 1\}$ is a 4-wise independent hash function.

We maintain these counters approximately using the classical Morris counters (Morris (1978)) on the positive and negative components. We decompose $A_{r,t}$ into positive and negative parts: $A_{r,t} = A_{r,t}^+ - A_{r,t}^-$. Note that $A_{r,t}^+ + A_{r,t}^- = \sum_j y_j = \|x\|_1 = m$.

We use Morris counters for $A_{r,t}^+$ and $A_{r,t}^-$ with an error parameter $\varepsilon'$. Let $\hat{A}_{r,t}$ be the resulting estimate of $A_{r,t}$. The absolute error is bounded by: $|\hat{A}_{r,t} - A_{r,t}| \le \varepsilon'(A_{r,t}^+ + A_{r,t}^-) \le \varepsilon' m$.

Let $\hat{Y}$ be the final AMS estimator computed using the approximate counters $\hat{A}_{r,t}$. We analyze the error introduced by the approximation in a single estimator. Let $Y' = \frac{1}{R} \sum_r A_r^2$ and $\hat{Y}' = \frac{1}{R} \sum_r \hat{A}_r^2$.

The error in the squared estimate is $|\hat{A}_r^2 - A_r^2| = |\hat{A}_r - A_r||\hat{A}_r + A_r|$. Since $|A_r| \le m$ and $|\hat{A}_r - A_r| \le \varepsilon' m$, we have $|\hat{A}_r| \le m(1 + \varepsilon')$. Thus, $|\hat{A}_r^2 - A_r^2| \le (\varepsilon' m)(2m + \varepsilon' m) = O(\varepsilon' m^2)$.

The total error due to approximation is $|\hat{Y}' - Y'| \le O(\varepsilon' m^2)$. We require this error to be small relative to the quantity being estimated: $|\hat{Y}' - Y'| \le \frac{\varepsilon}{4} \|y\|_2^2$.

Since $x$ is $k$-sparse and $\|x\|_1 = m$, by Cauchy-Schwarz, $\|x\|_2^2 \ge m^2/k$. Since $\|y\|_2^2 \approx \|x\|_2^2$, we have $\|y\|_2^2 \ge \Omega(m^2/k)$ with good probability.

We set $\varepsilon'$ such that $O(\varepsilon' m^2) \le \frac{\varepsilon}{4} \Omega(m^2/k)$. This requires $\varepsilon' = O(\varepsilon/k)$.

**Space Complexity** The space required for a Morris counter with error parameter $\varepsilon' = O(\varepsilon/k)$ up to a maximum count $m$ is $O(\log(1/\varepsilon') + \log\log m) = O(\log(k/\varepsilon) + \log\log m)$ bits.

We have $2RT = O(\frac{1}{\varepsilon^2} \log(1/\delta))$ such counters in total. The total space for the counters is:

$$O\left( \frac{1}{\varepsilon^2} \log\left(\frac{1}{\delta}\right) \left(\log\left(\frac{k}{\varepsilon}\right) + \log\log m\right) \right).$$

The space for the hash function $h : [n] \to [M]$ is $O(\log n)$. The space for the AMS hash functions is subsumed by the counter space.

The total space complexity is:

$$O\left( \frac{1}{\varepsilon^2} \log\left(\frac{1}{\delta}\right) \left(\log\left(\frac{k}{\varepsilon}\right) + \log\log m\right) + \log n \right).$$

This achieves the desired complexity where both $n$ and $k$ appear only in logarithmic factors relative to the $1/\varepsilon^2$ term. $\qquad\square$

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

## 4 LLM USE

LLMs were used to polish the writing in the introduction, as well as to expand proof outlines. All arguments were subsequently edited by hand.

