# OpenReview forum: "High Probability Streaming Lower Bounds for $F_2$ Estimation"
_ICLR.cc/2026/Conference — ICLR 2026 Conference Withdrawn Submission_

### Official Review · Reviewer_Vduz · 2025-10-23

**Soundness:** 1
**Presentation:** 2
**Contribution:** 3
**Rating:** 4
**Confidence:** 4

**Summary:**

The paper studies the streaming lower bound for estimating $F_2$ in the insertion-only model. It claims that any one-pass streaming algorithm achieving an $\epsilon$-additive error with failure probability $\delta$ requires $\Omega(\log \frac{\epsilon\sqrt{n}}{\log(1/\delta)}\cdot\frac{1}{\epsilon^2}\cdot \log\frac{1}{\delta})$ size of memory. Conceptually, this extends the constant-$\delta$ tight lower bound of Braverman-Zamir (2024) to regime where $\log(1/\delta)\leq \sqrt{n}\epsilon$. Technically, this paper follows from Braverman-Zamir framework but replaces "Exam Set Disjointness" with an "Exam Mostly Set Disjointness" variant, inspired by Kamath–Price–Woodruff (2021).

**Strengths:**

Tight streaming lower bound for estimating $F_2$ is very important and notoriously challenging; pushing from constant $\delta$ to very small $\delta$ is a meaningful target. Even if largely building on Braverman-Zamir (2024), a clean and correct extension to small $\delta$ would be of interest to the streaming algorithm community.

**Weaknesses:**

My main concern is the proof soundness and the presentation clarify. Many proof steps are stated without sufficient detail to verify. For example, in the proof of Lemma 2, the authors write "Using the chain rule for mutual information and the derivation involving D as in the proof of Lemma 4.3 in Braverman-Zamir (2024), we have", but without providing further details about Lemma 4.3. A detailed revision is needed to make the argument self-contained and checkable.

Besides, the writing has plenty of issues.  Specific examples:

1. The citation commands \citep and \citet are not used appropriately.
2. The reference should be updated (e.g. Braverman-Zamir (2024)) and more comprehensive. I suggest adding a brief survey of recent upper and lower bounds for $F_2$ estimation across various streaming models (insertion-only, turnstile, random-order, multi-pass).
3. Theorems 2 and 3 are identical; so are Definitions 1 and 2. They should be merged and restated once.
4. In Definition 1, line 91: should "an instance of case (i) with" be "an instance of case (ii) with"?
5. Line 95: "In n order to" => "In order to".
6. Line 135: "$1n-\delta$" => "$1-\delta$".
7. $M$ is used both for the memory state (line 144) and for the overlap parameter (line 148); these should be separated.
8. In the displayed equation in Line 186: $D$ => $P$?
9. Proposition 1: Clarify how the parameter $p$ enters the bound, and specify whether the random sample is performed with or without replacement.

**Questions:**

Could the approach be adapted to the random-order insertion-only setting? Can it be extended to the multi-pass setting?

---

### Official Review · Reviewer_WDpk · 2025-10-29

**Soundness:** 2
**Presentation:** 2
**Contribution:** 3
**Rating:** 4
**Confidence:** 4

**Summary:**

Consider the following problem.
Suppose we are in the streaming setting.
Let $U$ be the universe of size $n$ and $x_i$ be the number of occurrence of $i$ in the data stream for $i\in U$.
We would like to estimate the second frequency moment which is defined as $\sum_{i\in U} x_i^2$ within a $(1+\epsilon)$ multiplicative factor with probability at least $1-\delta$.
Our goal is to minimize the space complexity for achieving this estimation.
The celebrated AMS sketch gives an upper bound of $\frac{1}{\epsilon^2}\cdot \log (\frac{1}{\delta}) \cdot \log n$ space.
On the other hand, a recent result by Braverman and Zamir showed the lower bound of $\frac{1}{\epsilon^2}\cdot \log n$ space for constant $\delta$.
Therefore, the regime of dependence on $\delta$ is still missing and the paper provides the tight lower bound of $\frac{1}{\epsilon^2}\cdot \log (\frac{1}{\delta}) \cdot \log n$ space.

The main idea is to reduce the problem from Exam Mostly Set Disjointness (EMostlyDISJ) which is defined as follows.
Suppose there are $t$ players and one referee.
Let $U$ be the universe.
Each player $i$ has a subset $S_i$ of $U$ and the referee has an element $j$ of $U$.
It is guaranteed that $S_1,\dots, S_t$ satisfy one of the two cases: (i) $S_1,\dots,S_t$ are $M$-almost-disjoint or (ii) there is a unique element $j_0$ in $U$ that is common to at least $ct$ of the sets for some constant $c \in (0,1)$.
The communication is one-way that only player $i$ can send messages to player $i+1$ for $i=1,\dots, t-1$ and player $t$ can send messages to the referee.
With failure probability $\delta$, the referee needs to decide whether the sets $S_1,\dots,S_t$ satisfy (i) or (ii) and if (ii) then decide whether $j_0 = j$.
Finally, the authors prove that the lower bound of this communication game to be $\frac{1}{\epsilon^2}\cdot \log (\frac{1}{\delta}) \cdot \log n$.

Additionally, the authors give the results for different variations of the problem.

**Strengths:**

- The paper studies a classical data streaming problem and this allows readers to have a chance of visiting traditional perspectives of learning theory.

- The result provides a tight lower bound for a classical data streaming problem.
It provides new insights into the fundamental limitations.

**Weaknesses:**

- The paper may need a good amount of work to improve its presentation.
Judging from the proofs, it seems multiple lemmas are from the previous result and it may need some work to first introduce the idea from the previous work.
I still have a hard time on understanding the reduction from EMostlyDISJ to the second frequency moment problem.
It may be helpful to describe the reduction more rigorously.

**Questions:**

- Theorem 1, 2 and 3: I think they are the same.
There is a typo $1n-\delta$.

- Definition 1 and 2: I think they are the same.
Should it be ``... the referee must decide if the input is an instance of case (ii) ...'' because case (ii) is the case where there is a unique common $j_0$?
It may be helpful to move the definition of M-almost-disjoint in line 147 before Definition 1.

- Line 100: What is a super-item?

- Line 102: ``... more than in the NO instance ...''

- Line 160-161: It may be helpful to define the $F_{ct,t}$ problem more rigorously.

- Lemma 1: There is an extra . at the beginning of the lemma.

- Lemma 2: There is a missing . at the end of the first line.

---

### Official Review · Reviewer_czhm · 2025-10-31

**Soundness:** 3
**Presentation:** 2
**Contribution:** 2
**Rating:** 4
**Confidence:** 4

**Summary:**

The paper studies the classical problem of estimating the second frequency moment $F_{2}=\sum_{i=1}^n f_i^2$ in the insertion-only streaming model, where $f_i$ is the frequency of element $i$ in the stream. The well-known AMS algorithm, an $(\varepsilon, \delta)$ algorithm for estimating $F_2$, has a space complexity of $O({1\over \varepsilon^2}\cdot \log n \cdot \log ({1\over \delta})$. In a recent work, Braverman--Zamir (2024) showed that this indeed is tight in general with respect to $n$ and $\varepsilon$:  for constant failure probability $\delta$, any such algorithm will require $\Omega({\log n \over \varepsilon^2})$ space. However, in Braverman--Zamir (2024), the dependency of failure probability $\delta$ on space complexity is missing. The current paper builds on this work to obtain *tight dependence on the failure probability* $\delta$ in the lower bound. They show that for $\delta \geq {1\over 2^{\varepsilon \sqrt{n}}}$, $\log({1\over \delta})$ multiplicative factor in the space complexity is necessary. For example if $\delta = 1/n$ (a natural setting), the lower bound matches the upper bound asymptomatically in all three parameters.
They also identify special regimes (bounded-frequency streams and sparse streams) in which the general lower bound does not apply, and give algorithms with improved space complexity.

**Strengths:**

The complexity of $F_{2}$ estimation is fundamental in data streaming. Precisely characterizing dependence on all parameters is valuable. So the results are definitely worth publishing and is useful to know for those working in streaming algorithms theory. Technically the paper appears good (although I did not check the proofs carefully due to time constraint to know how far it is different from earlier work).

**Weaknesses:**

The result, while technically strong and interesting, is highly specialized. It may appeal mainly to researchers in streaming complexity and communication complexity, making it somewhat narrow for ICLR’s broader audience of machine learning researchers. The paper would benefit from significant proofreading. There are several typographical and formatting issues. Examples: in the statement of the main theorem (Theorem 2, which repeats Theorem 1 -- what is $1n-\delta$?). There is mention of $\delta$ in the abstract without defining it.
Theorems 1,2, and 3 are all the same, why have different numberings?

**Questions:**

It will be nice to explicitly state what regimes do the lower bounds match the known upper bounds? Can this be stated explicitly in a discussion after theorem statement? Could you provide some open problems arising from this line of work? For instance, are there natural cases where the gap between lower and upper bounds remains for other moment estimation? A related work section will be useful to understand the context of your work.

---

### Official Review · Reviewer_zBJb · 2025-11-01

**Soundness:** 4
**Presentation:** 2
**Contribution:** 3
**Rating:** 4
**Confidence:** 3

**Summary:**

This work studies the space complexity of estimating $F_2$ in the data stream model. The main contribution is an optimal lowerbound on the space complexity: $ \Omega(1/\epsilon^2\log n \log 1/\delta)$, where $\epsilon$ is the approximation error and $\delta$ is the probability error.
Earlier work established a lowerbound of $ \Omega(1/\epsilon^2\log n)$, the main contribution of this work is to improve this to a bound that includes $\delta$ parameter.

**Strengths:**

Estimating $F_2$ over data streams has been a well-studied problem, at least for the past 3 decades. The known upperbound is $O(1/\epsilon^2\log n \log 1/\delta). Thus, this work settles the space complexity of this problem.

**Weaknesses:**

The main weakness is the presentation and over-reliance on the proof of BravermanZamir24. It is nearly impossible to follow the proof and details unless one is completely familiar with the work of BravermanZamir24. I understand that the authors had to strike a balance between repeating the claims and technicalities of BravermanZamir24 and conveying their contributions. But in the current form, it is difficult to appreciate the work. There are many phrases/notions that are left undefined, and the reader is forced to infer what they mean. In my view, the paper is not ready to be published, in the current form,

**Questions:**

Few questions/concerns
1. Line 158-1599: This seems to be a critical ingredient of the proof and yet is stated very informally.
2. Lemma 1: The notion "No-instance distribution" is not introduced earlier, though one can infer what it means.
3. Proposition  1.  Not sure what "as above" if refering to. What is $t$ ?
4. Line 227: What is the choice for |U|?
5. Corollary: What is "total set size", I assume it is the size of all sets together?
6. Line 255: What is super-element size?
7. As far as I can see, the proof relies on the distribution $\mu_p$. So, what role does the hard distribution from definition 3 play?
8. At many places, the authors state "as in proof of BravermanZamir24". Is it possible to give few more details?

---

### Note · Authors · 2026-01-15

I have read and agree with the venue's withdrawal policy on behalf of myself and my co-authors.